# Metabolomic-Based Studies of the Intake of Virgin Olive Oil: A Comprehensive Review

**DOI:** 10.3390/metabo13040472

**Published:** 2023-03-25

**Authors:** Alejandra Vazquez-Aguilar, Estefania Sanchez-Rodriguez, Celia Rodriguez-Perez, Oscar Daniel Rangel-Huerta, Maria D. Mesa

**Affiliations:** 1Department of Biochemistry and Molecular Biology II, University of Granada, Campus Cartuja s/n, 18071 Granada, Spain; 2Institute of Nutrition and Food Technology “José Mataix”, Biomedical Research Center, University of Granada, Parque Tecnológico de la Salud, Avenida del Conocimiento s/n, 18016 Granada, Spain; 3Instituto de Investigación Biosanitaria de Granada ibs, 18012 Granada, Spain; 4Department of Nutrition and Food Science, University of Granada, Campus Melilla C/Santander, 52005 Melilla, Spain; 5Section of Chemistry and Toxinology, Norwegian Veterinary Institute, P.O. Box 64, N-1431 Ås, Norway; 6Primary Care Promotion of Maternal, Child and Women’s Health for Prevention of Adult Chronic Diseases Network (RD21/0012/0008), Institute of Health Carlos III, 28029 Madrid, Spain

**Keywords:** olive oil, metabolomics, phenolic compounds, hydroxytyrosol, pentacyclic triterpenes

## Abstract

Virgin olive oil (VOO) is a high-value product from the Mediterranean diet. Some health and nutritional benefits have been associated with its consumption, not only because of its monounsaturated-rich triacylglycerols but also due to its minor bioactive components. The search for specific metabolites related to VOO consumption may provide valuable information to identify the specific bioactive components and to understand possible molecular and metabolic mechanisms implicated in those health effects. In this regard, metabolomics, considered a key analytical tool in nutritional studies, offers a better understanding of the regulatory functions of food components on human nutrition, well-being, and health. For that reason, the aim of the present review is to summarize the available scientific evidence related to the metabolic effects of VOO or its minor bioactive compounds in human, animal, and in vitro studies using metabolomics approaches.

## 1. Introduction

Virgin olive oil (VOO) is the most important fatty source in the Mediterranean diet (MedDiet), which is one of the healthiest diets worldwide. VOO contains a saponifiable fraction made up of triacylglycerols (TAG; 97–99%), with oleic acid (C18:1n9) as the main fatty acid (68–81.5%). A systematic review and meta-analysis focused on the consumption of monounsaturated fatty acids (MUFA), and their relationship with cardiovascular disease (CVD) concluded an overall risk reduction of all-cause mortality (11%), cardiovascular mortality (12%), cardiovascular events (9%), and stroke (17%) [1]. The presence of these MUFA in cell membranes confers stability from oxidative damage and improves their fluidity and functions [2]. Besides, VOO contains 2% of non-saponifiable minor components, including phenylalcohols, secoiridoids, pentacyclic triterpenes, sterols, and tocopherols, among others. These compounds are responsible for many of the VOO health benefits, and within them, secoiridoids are responsible for the organoleptic properties [3,4]. The major phenolic compounds characterized in VOO are hydroxytyrosol, tyrosol, and the secoiridoids oleuropein aglycon, ligstroside aglycon, oleocanthal and oleacein [5]. Antioxidant properties have been attributed to these compounds. Hydroxytyrosol exerts antioxidant properties [6], improves endothelial function, reduces the expression of cell adhesion molecules, increases the availability of nitric oxide, and neutralizes intracellular free radicals [7]. It also has demonstrated selective toxicity against cancer cells, inducing apoptosis and protecting non-tumorigenic cells [8]. Pentacyclic triterpenes, mainly oleanolic and maslinic acids, are present in moderate amounts in VOOs [9]. Several studies reported that they are bioavailable in humans [10,11] and have demonstrated vasoprotective [11], metabolic [12], antioxidant and anti-inflammatory benefits in humans [13] and obese mice [14]. In addition, secoiridoids have also demonstrated beneficial properties. Oleuropein aglycon has demonstrated antioxidant capacity [5], metal-chelating and free radical scavenging activities [15] and anti-tumor effects [5]. Oleocanthal is a potent anti-inflammatory [5], antioxidant, neuroprotective [15] and antiproliferative molecule [16]. VOO also contains oleacein, which has recently shown a protective effect on experimental autoimmune encephalomyelitis and may normalize gut alterations associated with the disease [17], and ligstroside aglycons that reduces cell proliferation and increases cell death of liver cancer cells [16]. However, the molecular mechanism implicated in all these beneficial effects remains unknown. 

Metabolomics is the science that studies low molecular weight chemical compounds (<1500 Da) existing in biofluids, biological tissues or cells as a consequence of genetic, metabolic, physiological, or pathological conditions [18]. The metabolome represents the final step in a biological system, and metabolites are the final functional entities that can inform about the physiological or pathological phenotypes [19]. They provide information about what we eat by describing new dietary biomarkers that could identify dietary exposures with a high level of detail and precision, and also the metabolic pathways that might explain the beneficial, healthy effects attributed to specific food components at a molecular level [20]. Traditionally, two main approaches are used in metabolomics analysis: untargeted and targeted. Untargeted metabolomics analysis focuses on the determination of as many metabolites as possible, aiming at the coverage rather that the quantification. Targeted metabolomics analysis is built on prior knowledge and relies on the determination and quantification of specific metabolites of interest. 

The most employed analytical techniques are liquid and gas chromatography (LC and GC, respectively), coupled with mass spectrometry (MS) and nuclear magnetic resonance (NMR). By using high-resolution MS (HRMS) coupled with diverse sample preparation steps, the broad chemical complexity of the metabolome can be assessed. Indeed, a combination of separative methods is needed for in-depth investigations [21,22]. LC is perhaps the most popular method because of its high sensitivity, availability and versatility, providing wide coverage of the metabolome [21,23]. In recent years, the LC-MS has highlighted its enormous potential because of the minimal requirement of samples, simple pretreatments, and the ability to analyze samples in their natural state [24]. In addition, it has the advantage that it rarely requires derivatization steps and, hence, is quicker, relatively easier to perform and less expensive; it only requires deproteinization with different polar solvents depending on the sample [25]. GC-MS is the most effective for the analysis of the volatile fraction of samples, mostly composed of non-polar molecules. On the other hand, NMR has been used for over 40 years to perform metabolomic analyses in diverse biofluids and tissues. It is characterized by high levels of robustness and reproducibility, instrument stability, uncomplicated sample preparation, strong quantitative character, non-destructive nature, and easy automation. However, the big size of the instrument, the expensive maintenance and its low sensitivity, especially compared to mass spectrometry, are their main limitation [21].

Despite the well-described beneficial effects associated with VOO consumption, there is still a lack of information on the metabolomic changes induced after VOO intake, alone or enriched in bioactive compounds. The aim of the present review is to compile the scientific evidence related to the metabolic effects of VOO and its minor bioactive compounds using metabolomics approaches in human, animal, and in vitro studies.

## 2. Materials and Methods

The search was conducted in Medline through PubMed (US National Library of Medicine National Institutes of Health) and SCOPUS using the following research equation: “olive oil” AND “metabolomics.” We conducted the search from the beginning of the literature until November 2022. Studies included in this review met the following inclusion criteria: (1) human or animal studies that used metabolomics for evaluating the effects of olive oil consumption (non-modified or enriched with bioactive compounds) or the isolated bioactive components of olive oil, (2) in vitro cellular studies analyzing the molecular mechanisms of olive oil or any of its components. The exclusion criteria were: (1) direct metabolomic analyses of the oils and (2) reviews. 

The search yielded 88 results from PubMed and 146 from SCOPUS. After eliminating duplicated articles, titles and abstracts were screened to determine whether they met the inclusion criteria. In case of doubt, the full text was evaluated for further consideration. One hundred twenty-four papers did not meet the inclusion criteria and were excluded. Finally, 22 papers were selected: 12 were related to human studies, 7 were aimed at animal studies, and 3 were focused on in vitro experiments. 

## 3. Results

Table 1 and Table 2 include information about human clinical interventions, sustained or postprandial, respectively, carried out with olive oils or polyphenol-enriched olive oils. They include (1) the metabolites identified after the olive oil consumption compared with the control intervention unless otherwise indicated; (2) the level of identification based on the classification proposed by Schymanski et al. (2014), which established level 1 for confirmed structure by reference standard, level 2 for probable structure and level 3 for tentative candidate [26]; and (3) the metabolomics conclusions. All studies are ordered chronologically in tables. The same information is provided in Table 3 for animal studies and in Table 4 for in vitro experimental studies after the consumption of olive oils or hydroxytyrosol obtained from the olive fruit.

Down-regulated: phospholipids and their degradation products (lysoPC (C20:4), palmitoleic acid (C16:cis [9]1), PC (C16:0, C20:4), PC (C16:1, C18:2) and sphingomyelin (d18:1, C24:0)

### 3.1. Metabolomics Approaches in Humans

Twelve papers describing metabolites after olive oil, VOO, and extra virgin olive oil (EVOO) intake in humans were found (Table 1 and Table 2). Ten studies identified metabolites related to a beneficial effect, while three studies identified metabolites derived from olive oil intake. The Prevention with Mediterranean diet (PREDIMED) study is the first clinical trial demonstrating that dietary intervention with MedDiet supplemented with EVOO may decrease the morbidity and mortality due to CVD in high CVD-risk adults [49]. In that study, the goal was to consume daily 50 g or more of a polyphenol-rich EVOO. Five metabolomics analyses have been published from different sub-cohorts of the PREDIMED study. Three PREDIMED sub-studies compared plasma metabolites of a group of 230 subjects that suffered cardiovascular events compared with more than 780 participants without cardiovascular accidents, both before and after 1-year of intervention, using LC targeted-metabolomics approaches [28,29,31]. The first work [29] analyzed phosphatidylcholines, phosphatidylethanolamines, ceramides, cholesterol esters, diacylglycerols, and TAG. The authors described a baseline lipidomic plasma profile associated with CVD risk and future cardiovascular events, concretely lipid metabolites with a longer acyl chain and higher number of double bonds. However, besides the decrease in cardiovascular events observed after 1-year consumption of MedDiet supplemented with EVOO, the authors did not find any significant association between the described metabolites and cardiovascular risk [29]. It was suggested that 1 year was not enough to detect measurable changes in these types of metabolites, and other different metabolites and mechanisms may have accounted for the observed clinical benefits [29]. Indeed, the second work [28] identified ceramides as plasma metabolites related to EVOO consumption and CVD prevention after 7.4 years of supplementation. Participants with a high risk of CVD presented higher plasma amounts of these sphingolipids, increased levels of total cholesterol, LDL, TAG, and diastolic blood pressure, and the ceramide score was associated with a 2.18-fold higher risk of CVD. The authors concluded that after the MedDiet intervention for 7.4 years with at least 50 mL/d of EVOO, these ceramides decreased, potentially modulating cardiovascular risk [28]. 

The third sub-study described plasma metabolites related to the tryptophan-kynurenine pathway and their relation to CVD and the consumption of EVOO for 1 year in a MedDiet context, demonstrating that a higher plasma concentration of tryptophan and lower kynurenine-related metabolites were associated with a decreased risk of CVD [31]. In fact, this is the only work that identifies tryptophan as a metabolite related to cardiovascular risk at a level I of identification. However, it cannot be concluded whether the beneficial effect is related to MedDiet itself or to the intake of EVOO. Therefore, further specific clinical trials aimed at the evaluation of EVOO are necessary to ascertain if its intake may modulate processes associated with changes in plasma tryptophan. 

Another PREDIMED sub-study analyzed metabolites related to the metabolism of carbohydrates, amino acids, lipids, and microbial cometabolites, within others, in a sub-cohort of nondiabetic participants. This study identified some urine metabolites related to CVD prevention and the consumption of EVOO using an untargeted NMR analysis after 1 year and a targeted approach after 3 years, compared with the baseline. Differences were observed for all the metabolites excreted after EVOO consumption at 1 and 3 years but creatinine at 1 year [27]. 

Finally, the last PREDIMED sub-study included 251 type 2 diabetes mellitus (T2DM) participants compared with 638 non-diabetic controls [32]. These authors analyzed circulating plasma concentrations of several glycolysis/gluconeogenesis and tricarboxylic acids cycle-related metabolites at baseline and after 1 year using an LC-targeted approach. Baseline presence of hexose monophosphate, pyruvate, lactate, alanine, glycerol-3 phosphate and isocitrate were associated with a higher risk of T2DM. After 1 year of intervention with a controlled low-fat diet, citrate, isocitrate and malate were associated with a higher risk of T2DM, whereas MedDiet plus EVOO tended (*p* = 0.071) to improve the evolution of those T2DM risk-associated metabolites [32]. Once again, it cannot be concluded if the effect was related to the MedDiet or EVOO, confirming the need for new clinical trials exclusively focusing on EVOO metabolomic modifications. 

The VOO and HDL functionality (VOHF) study was an intervention trial involving 33 hypercholesterolemic subjects supplemented with 25 mL/d for 3 weeks of (1) a control VOO with 80 ppm of phenolic compounds, (2) a phenolic-enriched VOO with 500 ppm of phenolic compounds (mainly secoiridoids), or (3) a phenolic-rich VOO with 250 ppm of phenolic compounds and enriched with 250 ppm of additional phenolic compounds from thyme (mainly flavonoids). Two metabolomics analyses from the VOHF study have been published employing NMR-targeted approaches (Table 1). One of them described the impact of VOO consumption on the TAG-HDL profile [34]. An increase in TAG containing MUFAs and a decrease in those containing PUFAs or SFAs were reported after the supplementation with the control low-polyphenol olive oil and the phenolic-enriched VOO, but not in the thyme polyphenol-enriched oil, indicating that the effect on HDL-TAG depends on the type of phenolic compounds, i.e., secoiridoids vs. flavonoids, and not on the VOO matrix. The assessment of the HDL lipidome is a valuable approach to identifying and characterizing new biomarkers of HDL functionality. Although TAG is a minor component of HDL, the observed changes in these particles drive HDL functionality toward a cardioprotective pattern [34]. In the second VOHF work, the intervention with the two phenol-rich VOO for 3 weeks modified metabolite excretion compared to the low-polyphenols VOO, indicating a favorable shift in the circulating metabolic phenotype associated with cardiometabolic diseases [35].

In 2017, a postprandial clinical trial in healthy volunteers compared plasma metabolomic profile at baseline and 2 h after the consumption of 40 mL of EVOOs rich in oleocanthal, a hydroxytyrosol-derived phenolic compound that has demonstrated ex vivo platelet anti-aggregation properties [37]. The study used a GC-targeted metabolomic approach to discriminate two different phenotypes: responders vs. non-responders to the EVOO intervention. Responders to the anti-aggregating effect of EVOO tended to have higher plasma concentrations of glucose and other monosaccharides and their corresponding acids, whereas non-responder volunteers had higher circulating citric acid cycle metabolites (malic, isocitric and citric acids) and non-esterified fatty acids (oleic acid). This study demonstrated that subjects with different metabolomics profiles had different platelet anti-aggregating responses after EVOO consumption [37]. 

Another randomized controlled trial in prediabetic subjects evaluated the consumption of a MUFA diet based on olive oil and a control diet for 12 weeks on plasma TAG and non-esterified fatty acids such as oleic acid, linoleic acid, palmitoleic acid, linolenic acid, eicosapentaenoic acid, docosahexaenoic acid, palmitic acid, arachidonic acid, and myristic acid using an LC-targeted approach. Despite the results showing that the MUFA diet decreased liver fat and increased hepatic and total insulin sensitivity, no different metabolites were found between groups for TAG fatty acids determined by LC-MS [30]. On the contrary, the modification of plasma TAG composition after an MUFA-rich diet has been described previously [50]; however, it should be considered that plasma fatty acids profile is usually determined by GC, not LC; therefore, differences in these methodologies may have influenced these results. 

The last clinical trial involved 10 healthy participants taking 80 g/d of VOO for one month. The LC-targeted approach revealed plasma alterations in several metabolic pathways modulated by VOO consumption, including the homeostasis of amino acids, one-carbon metabolism, and fatty acid oxidation [33]. These authors proposed a new method for the analysis of the exposome, defined as the cumulative measure of external agents and associated biological responses throughout the lifespan. Using a targeted approach, they quantified more than 1000 metabolites in urine and blood samples and identified twenty-two plasma and two urine metabolites after VOO supplementation (Table 1). This work validated hydroxytyrosol 3-sulfate and hydroxytyrosol 4-sulfate as biomarkers of olive oil intake [33]. Sulfated-derived metabolites of hydroxytyrosol, which are phase-II hepatic metabolites, have also been identified in plasma and urine after VOO intake by other authors [51,52]. The scientific data regarding the bioavailability of VOO polyphenols have been reviewed previously [52,53]. These studies have indicated that VOO simple phenols, such as hydroxytyrosol and tyrosol, are absorbed after ingestion, with efficiency from 75% to 100% that depends on the food matrix [54], and excreted as glucuronide conjugates in a dose-dependent manner [55]. These compounds have a significant metabolic and hepatic transformation, beginning in enterocytes and continuing in the liver [56]. Regarding secoiridoids, phase I of the metabolism implies the hydrolysis of their structure, producing an increase in phenyl alcohols, and metabolic phase II consists of the conjugation with glucuronic and sulfates [56]. VOO compounds are also biotransformed by gastrointestinal microbiota into different phenolic metabolites [57]. In addition, an in vitro experiment in Caco-2 cells has demonstrated the antioxidant properties of hydroxytyrosol and tyrosol-sulfated derivatives [58], suggesting that both the components present in the olive oil and their metabolites are responsible for the improvement of the antioxidant status after VOO intake, and may also be involved in the beneficial anti-inflammatory, anti-hypertensive and metabolic properties [59]. 

Two postprandial studies (Table 2) have concluded, using LC untargeted approaches, that the serum metabolome profile may discriminate the intake of VOO from other edible oils in obese [36] or healthy volunteers [38]. The first work compared the intake of four different breakfasts (muffins) prepared with four heated oils: (1) EVOO with 400 µg/mL of phenolic compounds, (2) sunflower oil, (3) sunflower oil enriched with 400 µg/mL of phenolic compounds from pomace oil, and (4) sunflower oil enriched with a synthetic antioxidant (400 µg/mL-dimethylsiloxane). It described differences in serum lipidic and aminoacidic metabolites (Table 2), in particular an increase in oxidation-derived fatty acids metabolites and changes in free fatty acids associated with different heated oils ingestion [36]. The other study showed that the postprandial metabolomic response to the consumption of various cooking fats: olive, soybean, palm, camellia oils and tallow, was related to bile acids and salts metabolites, fatty acid metabolism, and pyrimidine nucleosides, among others. In addition, oils with similar fatty acid composition, such as olive oil and camellia oil, showed different physiological responses [38], indicating that other minor compounds may influence the postprandial absorption and metabolism of oil components and in the subsequent metabolic response in vivo. 

Based on the available literature, it can be summarized that sustained consumption of VOO affects the metabolome and modifies metabolic pathways of carbohydrates, lipids, and amino acids. However, the employment of different metabolomic strategies involving several analytical methods and, mainly, differences in experimental designs make it difficult to identify specific metabolites associated with the beneficial effect of VOO supplementation.

### 3.2. Metabolomics Approaches in Experimental Studies

The bibliographic search yielded eight papers describing effects after the intake of VOO or its isolated components in animal studies (Table 3) and two describing in vitro experimental studies (Table 4). Within them, only one study identified biomarkers of intake. In 2011, Mellert et al. [39] identified discriminating metabolites in Wistar rats supplemented with 5 mL/kg of body weight/day of olive oil (65–85% oleic acid) at baseline and after 7, 14 and 28 days of intervention [39]. An untargeted metabolomics approach was employed using GC-MS and LC-MS/MS for the identification of metabolites related to lipid metabolism that was different in males and females. Some metabolites derived from the excess of fatty acids degradation, and others formed in the glycolysis process and used for the TAG synthesis were up-regulated after VOO consumption, whereas phospholipids and their degradation products were down-regulated compared with the control group [39]. 

Three works studied the metabolomic impact after ingestion of hydroxytyrosol as a supplement in a rat model of metabolic syndrome using an untargeted approach and focusing on the metabolic changes and their consequences [40,41,42]. In these studies, rats were supplemented with 20 mg/kg/day of hydroxytyrosol for 8 weeks, along with a high-fat and carbohydrate diet, to induce metabolic syndrome. The first study demonstrated a beneficial effect on adiposity, glucose and insulin tolerance, endothelial function, systolic blood pressure, left ventricular fibrosis and resultant diastolic stiffness, as well as in biomarkers of liver damage and oxidative stress compared with the control group without hydroxytyrosol supplementation. Using an LC-targeted metabolomic approach, the authors identified 24 plasma metabolites derived from hepatic or colonic microbiota metabolism, which potentially may be related to hydroxytyrosol supplementation [40]. In addition, they described that the excess of dietary fat and carbohydrates used to induce obesity was accompanied by lower plasma levels of six of these hydroxytyrosol metabolites compared with non-obese animals, which could be due to lower absorption, hepatic transformation or tissue accumulation [40]. When comparing the effect of hydroxytyrosol supplementation (20 mg/kg for 8 weeks) in the two obese groups of rats fed the high-carbohydrate and fat diet, they reported differences in 31 metabolites using two different analytical platforms: UPLC-Orbitrap and QqTOF [41]. Hydroxytyrosol supplementation induced down-regulation of 16 metabolites involved in the fatty acid biosynthesis, mainly unsaturated fatty acids, and in the metabolisms of linoleic acid, arachidonic acid, sphingolipid and retinol, whereas the glycerolipid metabolism was the main up-regulated metabolic pathway (Table 3). The QqTOF-based approach identified 12 endogenous metabolites that were different between the control and hydroxytyrosol-treated groups: 10 were down-regulated, and two were up-regulated. The authors studied the relation of all those 31 metabolites with metabolic syndrome consequences derived from insulin resistance, lipolysis, prostaglandins biosynthesis, sphingolipid pathway, and hepatic disease, which were improved after hydroxytyrosol intake. These findings contributed to the elucidation of metabolic, cardiovascular, and hepatic benefits attributed to hydroxytyrosol and VOO intake [41]. Other data from the same study focused on the hepatic metabolome and described the effect of hydroxytyrosol intake on liver functions, mainly on lipid metabolism, by the use of LC and NMR techniques [42]. The supplementation with 20 mg/kg/day of hydroxytyrosol seems to mobilize and up-regulate different lipidic classes in plasma, specifically branched fatty acid esters of hydroxyl-oleic acids (OAHSA), denoting a benefit for metabolic syndrome, in agreement with other studies [60]. In addition, reduced glucose plasma levels were also observed in hydroxytyrosol-treated rats, showing an improvement in insulin sensitivity and, therefore, in the metabolic syndrome evolution [42]. 

In 2022, a study carried out in rats with metabolic syndrome induced by a high-fructose, and high-fat diet evaluated the metabolic effect of the ad libitum consumption of EVOO for 12 weeks, focusing on the metabolic profile and the role of gut microbiota [45]. This study used LC untargeted approaches to identify differences in metabolomic profiles in feces and serum among different groups. They reported 12 potential biomarkers of EVOO intake in feces, mainly glycerophospholipids, amino acids, peptides and analogs, and fatty acids and derivatives, while six potential biomarkers were identified in serum samples, mainly amino acids, peptides and their analogs. Amino acids play important roles in various metabolic processes altered during obesity and related CVD, and other studies have suggested a direct association between branched-chain and aromatic amino acids and CVD [61,62]. The study concluded that EVOO supplementation mainly altered amino acids, peptides and their analogs in feces and serum and associated those changes with gut microbiota metabolic function [45]. 

On the other hand, Ruocco et al., 2022 [46] analyzed the plasmatic amino acid profile related to the turnover of proteins of mice fed ad libitum a high-polyphenol EVOO diet compared with mice fed a saturated fatty acids-rich diet for 16 weeks. They used LC untargeted to analyze metabolites in plasma and urine and suggested that dietary consumption of polyphenol-enriched EVOO improves metabolic parameters and circulating biomarkers of metabolic health, tending to decrease branched-chain and aromatic amino acids [46]. Further studies are needed to establish the effect of EVOO components on amino acid metabolism and its implication on cardiovascular health. 

Two studies have described the effect of olive oil (69% oleic acid) intake compared to perilla oil (56% of linolenic acid) and palm oil (78% of palmitic acid), during 8 weeks, in Chinese mitten crabs (Eriocheir sinensis). One work reported that crab fed the olive oil diet grew faster and had lower concentrations of hepatic glycogen, TAG, and oxidative stress biomarkers, while metabolites related to glycolysis and the tricarboxylic acid cycle, intermediate for valine and leucine synthesis, and intermediate for glutathione synthesis were up-regulated compared with the perilla oil [43]. The other work described differences when comparing olive oil vs. palm oil intake in five metabolic pathways, including alanine, aspartate and glutamate metabolism, lysine biosynthesis and degradation metabolism, arginine and proline metabolism, pyrimidine metabolism and propanoate metabolism [44]. These data are different from those obtained by Guasch–Ferré et al. (2020) in humans with T2DM [32], but we think that differences in human and crab metabolisms make it difficult to compare results.

In 2012, Fernandez–Arroyo et al. studied the antiproliferative and pro-apoptotic activities of polyphenol-EVOO extracts on adenocarcinoma cells (HT29 and SW480) to identify molecules responsible for these actions. Those authors incubated cells in the presence of two different doses (0.01% and 0.1%) of 14 different EVOO extracts for 24 h. Phenolic compounds and their metabolites were identified by an LC-targeted metabolomics approach in the cytoplasm and culture medium in EVOO-treated cells but not in non-treated cells. Within them, quercetin was the main compound found in the cytoplasm, followed by oleuropein and its derivative decarboxymethyl oleuropein aglycone (DOA). In addition, authors associated the presence of quercetin or oleuropein aglycone and its derivatives with the antiproliferative and pro-apoptotic effect [47] (Table 4).

Finally, an In vitro study that simulated gastrointestinal digestion of reported 64 compounds derived from In vitro digestion of five commercial EVOOs. A marked abundance of flavonoids (15 compounds), followed by cholesterol and spirostanol analogs (15 compounds), was described by an untargeted metabolomics approach using UHPLC-QTOF; 10 compounds were confirmed as the most discriminant compounds during the In vitro gastrointestinal digestion process [48] (Table 4). Another In vitro study simulating the gastrointestinal digestion of phenolic alcohols hydroxytyrosol and tyrosol during a constant 24 h colonic metabolism described metabolites formed during the stomach and small intestine digestion that impact their availability and metabolic fate. In addition, they reported that the colon microbiota degrades in a similar way, both tyrosol and hydroxytyrosol [63].

It is worth mentioning that metabolomic analyses, especially those targeted approaches in which the validation and quantification of specific metabolites are carried out, can be a useful additional tool for supporting health claims since they can predict health risks or evaluate dietary intake [21]. The increasing scientific evidence relating to functional ingredients and their health effect might also be interesting for stakeholders and food companies, which could benefit from the added value attributed to their products by the presence of the ingredient responsible for the claimed bioactivity [64]. The EFSA’s health claim criteria for functional foods require information about the substance (bioactive compound), the study of the physiological effects, and the estimation of the cause-effect relationship. In this last step, one of the main problems is that many of the studies use biomarkers that are not significantly reliable by the Agency. In this regard, the use of metabolomics could play an important role. However, taking into consideration that nutrimetabolomics is still a young science under development and that more standardized and well/designed studies are necessary [65], we are still far from the implementation of metabolomics as a routinary tool for health claims support. Therefore, at this moment, available data cannot support olive oil EFSA health claims.

## 4. Conclusions

The present review highlights the need for clinical studies necessary to understand the molecular and metabolic mechanisms of action of VOO components. Although metabolomics studies derived from the consumption of VOO and their metabolic routes are scarce, it has been shown that the intake of VOO causes an increase in derivatives of hydroxytyrosol oleuropein and oleic acid, such as phosphatide derivatives, which may be used as markers of VOO consumption. In addition, studies have identified possible metabolic pathways related to glycolysis, the tricarboxylic acid cycle, and amino acids metabolism that are modulated by VOO intake and, therefore, may be implicated in the benefits of this healthy oil. However, differences in analytical strategies, the heterogeneity of the experimental designs and interventions, the use of different oils and doses, the bioavailability of VOO bioactive compounds into different matrixes, and the fact that many of the studies carried out in humans evaluate the effect of VOO in the frame of Mediterranean diet, do not allow us to reach specific conclusions on particular metabolites and metabolic pathways affected by VOO intake.

## Figures and Tables

**Table 1 metabolites-13-00472-t001:** Results of the main human metabolomics studies.

Reference	Study Data		Main Metabolites Identified after Olive Oil Consumption	Conclusions
**Vázquez-Fresno et al., 2015** [27]	**Subjects**	N = 98 [53–79 years] nondiabetic at high CVD risk 70 females 28 males	Up-regulated after 1 y: creatinine, citrate, cis-aconitate Up-regulated after 3 y: creatinine and citrate	Some urine metabolites may discriminate dietary pattern
	**Intervention**	PREDIMED Study MedDiet + EVOO 1 year vs. 3 years		
	**Technique**	Untargeted NMR (1-year results) and targeted (3-year results) NMR		
	**Sample**	Urine		
	**MSI**	Level 2		
	**Statistical analysis**	Multivariate Unsupervised PCA Supervised OSC-PLS-DA		
**Wang et al., 2017** [28]	**Subjects**	N = 980 [55–80 years] high risk CVD 541 females, 476 males	Down-regulated: 4 ceramides (C16:0, C22:0, C24:0 and C24:1)	Positive association between ceramide and CVD risk MedDiet + EVOO may mitigate the potentially deleterious effects of elevated plasma ceramides
	**Intervention**	PREDIMED Study MedDiet + EVOO Baseline vs. 7,4 years		
	**Technique**	Targeted LC-MS		
	**Sample**	Plasma		
	**MSI**	Level 3		
	**Statistical analysis**	Multivariate		
**Toledo et al., 2017** [29]	**Subjects**	N = 983 [55–80 years] high risk CVD 541 females, 479 males	Up-regulated: lysoPE (22:6), PC plasmalogens (34:2), PE plasmalogens (36:1), ceramide (24:1), sphingomyelines (18:1, 18:0 and 24:1) Down-regulated: PC (36:4b), PC (38:4), PE (38:5 and 38:4), PE plasmalogens (36:5 and 38:5), cholesterol esters (16:1), diacylglycerols (32:0), TAG (42:0, 44:0, 46:0, 48:0 and 50:0) No significant differences were found in adjusting *p*-values for multiple comparisons	Baseline lipid metabolomic profile was associated with the risk of CVD and was reduced after sustained consumption of MedDiet + EVOO
	**Intervention**	PREDIMED Study MedDiet + EVOO Baseline vs. 1 year		
	**Technique**	Targeted UHPLC-Orbitrap MS		
	**Sample**	Plasma		
	**MSI**	Level 3		
	**Statistical analysis**	Multivariate		
**Errazuriz et al., 2017** [30]	**Subjects**	N = 43 [mean value 62 years] Prediabetics 19 females 25 males	TAG fatty acids composition and nonsterified fatty acids: oleic acids, linoleic acids, palmitoleic acids, linolenic acids, eicosapentaenoic acids, docosahexaenoic acids, palmitic acids, arachidonic acids, myristic acids, and TAG	No differences were found in the metabolites analyses in MUFA vs. control diet after 12 wk
	**Intervention**	MUFA diet (50% olive oil); fiber-rich diet; Control diet (high-carbohydrate, low-fat and low fiber) MUFA vs. fiber-rich vs. control diet		
	**Technique**	Targeted LC-MS		
	**Sample**	Plasma		
	**MSI**	Level 3		
	**Statistical analysis**	Univariate		
**Yu et al., 2017** [31]	**Subjects**	N = 985 [55–80 years] high risk CVD 529 females, 456 males	Up-regulated: tryptophan Down-regulated: kynurenine, kynurenic acid, 3-hydroxyanthranilic acid and quinolinic acid	Increases in plasma tryptophan after 1 y was inversely associated with incident CVD MedDiet + EVOO attenuated the deleterious effect of low levels of tryptophan
	**Intervention**	PREDIMED Study MedDiet + EVOO Baseline vs. 1 year		
	**Technique**	Targeted LC-MS		
	**Sample**	Plasma		
	**MSI**	Level 1		
	**Statistical analysis**	Multivariate		
**Guasch-Ferre et al., 2020** [32]	**Subjects**	N = 889 [55–80 years] high risk CVD and T2DM risk 573 females 369 males	Down-regulated: isocitrate and malate No significant interactions were found after adjusting for multiple comparisons	Glycolysis/gluconeogenesis and TCA-related metabolites panel positively associated with T2DM risk MedDiet + EVOO or nuts may counterattack the harmful effects of those metabolites
	**Intervention**	PREDIMED Study MedDiet + EVOO Baseline vs. 1 year		
	**Technique**	Targeted LC-MS		
	**Sample**	Plasma		
	**MSI**	Level 3		
	**Statistical analysis**	Multivariate		
**Gonzalez-Dominguez et al., 2020** [33] *****	**Subjects**	N = 10 healthy [Mean value: 40 years) 4 females 6 males	Up-regulated in urine: HT 3-sulfate and HT 4-sulfate Up-regulated in plasma: ethanolamine, urea, s-adenosylmethionine, dimethylglycine, pyroglutamic acid, asymmetric dimethylarginine, trimethylamine, glutaryl-L-carnitine, succinic acid, azelaic acid, leucine, acetyl-L-carnitine, valine, s-adenosylhomocysteine, lysine, methionine, threitol, creatinine, glycochenodeoxycholic acid 3-glucuronide, indoleacetic acid, docosatetraenoic acid, phenylalanine	HT is bioavailable, and its metabolites are excreted in urine after one month of VOO intervention. Ingestion of olive oil modified plasma metabolome
	**Intervention**	Olive oil 80 g/day Baseline vs. 1 month		
	**Technique**	Targeted UHPLC-QTRAP		
	**Sample**	Urine and plasma		
	**MSI**	Level 2		
	**Statistical analysis**	Univariate		
**Fernandez-Castillejo et al., 2021** [34]	**Subjects**	N = 33 [35–80 years] hypercholesterolaemic 14 females 19 males	Up-regulated: TAG(FA18:1), SM(FA22:1), TAG56:5(FA20:3), TAG 54:2(FA20:1), TAG 52:2(FA16:0), TAG 52:2(FA18:1), PC(FA18:1/FA18:1), SM(FA22:1), TAG54:2(FA18:1), TAG 56:4(FA20:2), TAG 54:4(FA20:3), TAG 56:4(FA20:3), TAG 50:3(FA14:1), TAG 52:1(FA18:1), TAG 54:2(FA16:0) and TAG 54:3(FA20:2) Down-regulated: CE(FA22:6), TAG56:8(FA18:2), TAG 51:4(FA18:2), TAG 51:4(FA15:0), TAG54:7(FA22:5), CE(FA22:6), TAG56:8(FA18:2), TAG51:5(FA18:3), TAG 50:4(FA18:2), TAG 52:4(FA18:2), TAG 52:4(FA16:0) and TAG 53:3(FA18:2)	VOO impacts the HDL lipidome, in particular TAG species, independently of polyphenol content
	**Intervention**	VOHF Study 25 mL/day for 3 weeks of: VOO (80 ppm of TPC); FVOO (500 ppm of TPC); FVOOT (250 ppm of VOO TPC + 250 ppm of thyme TPC). Baseline vs. 3 weeks		
	**Technique**	Targeted NMR		
	**Sample**	Serum		
	**MSI**	Level 2		
	**Statistical analysis**	Multivariate Unsupervised PCA Supervised OPLS-DA		
**isFarras et al., 2022** [35]	**Subjects**	N = 33 [35–80 years] hypercholesterolaemic 14 females, 19 males	Down-regulated: glutamine, histidine, DMA, creatine, creatinine, valine, isoleucine Metabolites identified after the consumption of VOO enriched in phenolic compounds vs. a standard VOO	Phenol-enriched olive oils favorably shift circulating metabolites associated with cardiometabolic diseases
	**Intervention**	VOHF Study 25 mL/day for 3 weeks of: VOO (80 ppm of TPC); FVOO (500 ppm of TPC); FVOOT (250 ppm of VOO TPC + 250 ppm of thyme TPC). Baseline vs. 3 weeks		
	**Technique**	Targeted NMR		
	**Sample**	Serum		
	**MSI**	Level 2		
	**Statistical analysis**	Multivariate supervised M-OPLS-DA, PLS, Machine learning		

* Indicates studies of olive oil intake metabolites. CE, cholesteryl esters; CVD, cardiovascular disease; DMA, dimethylamine; EVOO, extra virgin olive oil; FA, fatty acid; HDL, high-density lipoprotein; HT, hydroxytyrosol; LC, liquid chromatography; MedDiet, Mediterranean diet; M-OPLS-DA, multilevel orthogonal partial least squares discriminant analysis; MUFA, monounsaturated fatty acids; MS, mass spectrometry; MSI, Metabolomics Standards Initiative; NMR, nuclear magnetic resonance spectroscopy; OPLS, orthogonal partial least squares; OSC, orthogonal signal correction; PC, phosphatidylcholine; PCA, principal component analysis; PE, phosphatidylethanolamine; PLS, partial least squares; PLS-DA, partial least square-discriminant analysis; PREDIMED, Prevention with Mediterranean Diet; QTRAP, mass spectrometer with electrospray ionization source and hybrid triple quadrupole analyser; SM, sphingomyelin; T2DM, diabetes mellitus type 2; TAG, triacylglycerols; TCA, tricarboxylic acid; TPC, total phenolic compounds; UHPLC, ultra-high pressure liquid chromatography; VOO, virgin olive oil; VOHF, VOO and HDL functionality.

**Table 2 metabolites-13-00472-t002:** Results of the main human postprandial metabolomics studies.

Reference	Study Data		Main Metabolites Identified after Olive Oil Consumption	Conclusions
**Ferreiro-Vela et al., 2013** [36] *****	**Subjects**	N= 26 obese 17 females [48–70 years] 9 males [39–70 years]	Up-regulated after 2 h: 3-hydroxydecanoic acid, 3-oxooctadecanoic acid, octadecanedioic acid (12,13-DHOME, 9,10-DHOME), palmitoleic acid (palmitelaidic acid), eicosenoic acid, disaccharide, lysoPE(18:1(9Z)/0:0), lysoPE(18:1(11Z)/0:0) Down-regulated after 2 h: tryptophanol, 9,10-dihydroxyoctadecanoic acid, palmitic acid, 5′-methylthioadenosine, 3-methyladipic acid (pimelic acid) and L-tryptophan Up-regulated after 4 h: 3-hydroxydecanoic acid, 9,10-dihydroxyoctadecanoic acid, 3-oxooctadecanoic acid, octadecanedioic acid (12,13-DHOME, 9,10-DHOME), palmitoleic acid (palmitelaidic acid), palmitic acid, eicosenoic acid and disaccharide Down-regulated after 4 h: L-tyrosine Down-regulated after 4 h vs. 2 h: glucosamine	Serum metabolites may discriminate the intake of different oils and the postprandial phase
	**Intervention**	Postprandial: Baseline, 2 and 4 h after a breakfast including 0.45 mL of EVOO/kg of body weight (400 µg/mL of TPC)		
	**Technique**	Untargeted LC-TOF/MS		
	**Sample**	Serum		
	**MSI**	Level 3		
	**Statistical analysis**	Multivariate Supervised PLS-DA		
**Agrawal et al., 2017 ** [37]	**Subjects**	N = 9 [20–50 y] healthy males	Up-regulated in responders: glucose, xylose and pinitol (carbohydrates), glycolic acid, gluconic acid and threonic acid (sugar acids) Up-regulated in non-responders: oleic acid (free fatty acid), malic acid, isocitric acid and citric acid (citric acid cycle metabolites)	Plasma metabolomics profiles may discriminate platelet response to EVOO intake
	**Intervention**	Postprandial: Baseline vs. 2 h after the intake of 40 mL of three EVOO		
	**Technique**	Targeted GC-TOF		
	**Sample**	Plasma		
	**MSI**	Level 3		
	**Statistical analysis**	Multivariate Supervised PLS-DA		
**Wang et al., 2018** [38] *****	**Subjects**	N = 17 [20–50 years] healthy males	Up-regulated: glycochenodeoxycholic acid, deoxycholic acid and hyodeoxycholic acid (bile acids and salts), 3-hydroxybutyric acid (fatty acid metabolism), uridine (pyrimidine nucleosides), traumatic acid, 2-ethyl-2-hydroxybutyric acid and mandelic acid Down-regulated: 5′-methylthioadenosine	Different metabolic profiles were observed between MUFA and SFA oils
	**Intervention**	Postprandial: Baseline vs. 2 and 4 h after the intake of 54 g of olive oil		
	**Technique**	Untargeted UHPLC-MS/MS QTOF		
	**Sample**	Serum		
	**MSI**	Level 2		
	**Statistical analysis**	Multivariate Supervised SPLS-DA		

* Indicates studies of olive oil intake metabolites. DHOME, dihydroxyoctadecenoic acid; EVOO, extra virgin olive oil; GC-TOF, gas chromatography time-of-flight mass spectrometry; LC, liquid chromatography; MUFA, monounsaturated fatty acids; MS, mass spectrometry; MSI, Metabolomics Standards Initiative; PE, phosphatidylethanolamine; PLS-DA, partial least square-discriminant analysis; QTOF, quadrupole time-of-flight; SFA, saturated fatty acids; SPLS, sparse partial least squares; TOF/MS, time-of-flight-mass spectrometry; TPC, total phenolic compounds; UHPLC, ultra-high pressure liquid chromatography.

**Table 3 metabolites-13-00472-t003:** Results of the in vivo animals’ metabolomics studies.

Reference	Study Data		Main Metabolites Identified after Olive Oil and Its Minor Bioactive Components Consumption	Conclusions
**Mellert et al., 2011** [39]	**Animals**	N = 10 Wistar rats	Up-regulated: ketone bodies (2-, and 3-hydroxybutyrate, only in females) and glycerol-3-phosphate (male and female) Down-regulated: phospholipids and their degradation products (lysoPC (C20:4), palmitoleic acid (C16:cis [9]1), PC (C16:0, C20:4), PC (C16:1, C18:2) and sphingomyelin (d18:1, C24:0)	Lipid metabolism was modified by olive and corn oils in similar ways Lower levels of phospholipids are due to the lower food consumption
	**Intervention**	Olive oil (65–85% of oleic acid) 5 mL/kg of body weight/day Baseline vs. 28 days		
	**Technique**	Untargeted GC-MS and LC-MS/MS		
	**Sample**	Plasma		
	**MSI**	Level 2		
	**Statistical analysis**	Univariate		
**Poudyal et al., 2017** [40]	**Animals**	N = 48 metabolic syndrome rat model	Up-regulated: HT, HT double oxidation, HT 2-ethoxyl acid, HT glucuronidation, HT glutathione conjugation, HT sulfation *, HT acetylation *, HT N-acetylcysteine conjugation, HT acetylation + sulfation *, HT methylation (homovanillic alcohol), homovanillic alcohol first alcohol to aldehyde *, homovanillic alcohol sulfation, homovanillic alcohol methylation, homovanillic alcohol acetylation, homovanillic acid, homovanillic acid aromatic hydroxylation, homovanillic acid glucuronidation, homovanillic acid sulfation, homovanillic acid methylation, homovanillic acid acetylation *, homovanillic acid glycine conjugation (carboxylic acid), homovanillic acid hydroxylation + methylation, 3,4-diphenylacetic acid, 3,4-diphenylacetic acid glucuronidation *, 3,4-diphenylacetic acid glycine conjugation (carboxylic acid) * Indicates discriminant metabolites down-regulated for the obese group compared with the control group, both treated with HT	Cardioprotective effects of HT were observed by attenuation of metabolic risk factors
	**Intervention**	Group 1: corn starch; Group 2: corn starch + 20 mg HT/kg/day Group 3: HCHF Group 4: HCHF + 20 mg HT/kg/day Baseline vs. 8 weeks		
	**Technique**	Targeted UHPLC-HRMS		
	**Sample**	Plasma		
	**MSI**	Level 3		
	**Statistical analysis**	Univariate		
**Lemonaski et al., 2017** [41]	**Animals**	N = 16 metabolic syndrome rat model	UPLC-Orbitrap up-regulated: an unknown metabolite and 3-methoxy-4-hydroxyphenylacetaldehyde (primary amide (fatty acyls)) UPLC-Orbitrap down-regulated: octadecanamide, fatty acid ester, unsaturated fatty acid/C24 bile acid (sterol lipids)/w-3 polyunsaturated fatty acid ethyl ester, unsaturated fatty acid/C24 bile acid (sterol lipids), C24 bile acid (sterol lipids), 1-alkyl,2-acylglycerophosphocholines (glycerophospholipids), retinoid (prenol lipids), oleamide, monoacylglycerophosphocholine, 18-oxocortisol, diacylglycerophosphoinositol, 3beta-(3-methyl-butanoyloxy)-villanovane-13alpha,17-diol, 5-hydroperoxy-7-[3,5-epidioxy-2-(2-octenyl)-cyclopentyl]-6-heptenoic acid, C24 bile acid, diacylglycerophosphoinositol, sn-3-O-(geranylgeranyl)glycerol 1-phosphate QqTOF up-regulated: (glycerol and 3-(3-hydroxyphenyl)propanoic acid) QqTOF down-regulated: lauric acid, linoleic acid, oleic acid, stearic acid, 3,7-dihydroxycholan-24-oic acid, (3beta,5alpha)-4,4-dimethylcholesta-8,14,24-trien-3-ol, myristic acid, palmitelaidic acid, 11,14,17-eicosatrienoic acid/8,11,14-eicosatrienoic acid, arachidonic acid/cis-8,11,14,17-eicosatetraenoic acid	HT decreases the biosynthesis of fatty acids, mainly unsaturated, and the metabolism of linoleic acid, retinol, sphingolipids and arachidonic acid, whereas glycerolipid metabolism is up-regulated These metabolites regulation may explain the positive effect of HT in cardiovascular, liver and metabolic changes induced by high-carbohydrate, high-fat diet-fed rats
	**Intervention**	Control diet: HCHF Enriched diet: HCHF + 20 mg HT/kg/day Baseline vs. 8 weeks		
	**Technique**	Untargeted UPLC-Orbitrap and UPLC-QqTOF		
	**Sample**	Plasma		
	**MSI**	Level 3		
	**Statistical analysis**	Multivariate Supervised PCA Unsupervised PLS-DA and OPLS-DA		
**Dagla et al., 2018** [42]	**Animals**	N = 15 metabolic syndrome rat model	Up-regulated: 9-ή 12-OAHSA (oleic acid hydroxyl stearic acid), unsaturated lipid acids, PC (22:6) or diacylglycerol phosphoserine, PC (20:4), γ-glutamine amino acid, glycerol, glycerol and/or glycine, choline, leucine, isoleucine and/or leucine Down-regulated: glucose and/or mannose, glucose, glucose and/or betaine, glucose-mannose, glucose and/or O-phosphocholine and lactate	HT is effective towards the mobilization of lipids and up-regulates branched fatty acid esters of hydroxy oleic acids, denoting the alleviation of the metabolic syndrome
	**Intervention**	Control group: HCHF HT group: HCHF+ 20 mg HT/kg/day Baseline vs. 8 weeks		
	**Technique**	Untargeted UPLC-HRMS and NMR		
	**Sample**	Liver		
	**MSI**	Level 2		
	**Statistical analysis**	Multivariate Supervised PCA Unsupervised PLS-DA and OPLS-DA		
**Ma et al., 2017** [43]	**Animals**	N = 360 crabs	Up-regulated: pyruvic acid, succinic acid, lactose, L-malic acid, D-gliceric acid, threitol (related to glycolysis and tricarboxylic acid cycle), methionine, 2-keto-isovaleric acid (intermediate for valine and leucine synthesis) and 2-hydroxybutanoic acid (intermediate of ketogenic amino acids breakdown), 6-deoxy-D-glucose, 2-hydroxypyridine and 3-hydroxypropionic acid Down-regulated: glutaconic acid (intermediate of ketogenic amino acids breakdown)	Compared with perilla oil-fed crabs, olive oil increased the degradation of glucose and lipids to provide energy for growth
	**Intervention**	Olive oil (69% oleic acid) and perilla oil (56% linolenic acid) Baseline vs. 8 weeks		
	**Technique**	Untargeted GC-MS		
	**Sample**	Serum		
	**MSI**	Level 3		
	**Statistical analysis**	Multivariate Supervised PCA Unsupervised PLS-DA and OPLS-DA		
**Ma et al., 2018** [44]	**Animals**	N = 360 crabs	Up-regulated: hydroxylamine, 3-hydroxypropionic acid and 2-hydroxypyridine Down-regulated: lysine and citrulline	Compared with palm oil-fed crabs, olive oil provides more energy, lower lipid accumulation and oxidative stress, and improves intestinal microbiota Palmitic acid-enriched palm oil tended to increase protein degradation and lipid accumulation-induced lipotoxicity
	**Intervention**	Olive oil (69% oleic acid) and palm oil diet (78% of palmitic acid) Baseline vs. 8 weeks		
	**Technique**	Untargeted GC-MS		
	**Sample**	Serum		
	**MSI**	Level 3		
	**Statistical analysis**	MultivariateSupervised PCA Unsupervised PLS-DA and OPLS-DA		
**Zhi-hao et al., 2022** [45]	**Animals**	N = 48 metabolic syndrome rat model	Feces up-regulated: proline, valine, cytidine, glutathione (reduced; amino acids, peptides, and analogs), oleic acid and FA 18:0 + 2O + SO_4_ Feces down-regulated: PE alkenyl 16, PE alkenyl 18, PE 16, PC 15 (glycerophospholipids) FA 18:4 +1O and citrulline Serum up-regulated: alanine-isoleucine, leucine and oleic acid. Serum down-regulated: 3,5-dibromo-L-tyrosine, folic acid and cytidine 5’-diphosphocholine	Supplementation with both high-oleic acid peanut oil and EVOO reduces diet-induced metabolic syndrome. The major pathway implicated in these metabolic effects is the BCAAs biosynthesis pathway.
	**Intervention**	Normal, HFHF, HFHF diet containing high-oleic acid peanut oil, HFHF containing EVOO. Baseline vs. 12 weeks		
	**Technique**	Untargeted UPLC-Q/TOF-MS		
	**Sample**	Feces and serum		
	**MSI**	Level 2		
	**Statistical analysis**	Multivariate Supervised PLS-DA		
**Ruocco et al., 2022** [46]	**Animals**	N = 19 C57BL/6N mice	Plasma down-regulated: proline Urine up-regulated: tyrosol-sulfate, HT, HT-sulfate, HT-acetate-glucuronide, homovanillic acid-glucuronide, oleuropein aglycone, ligstroside Significant differences could not be calculated for oleuropein and oleuropein aglycone-glucuronide because these compounds were non-detected in the SFA group	The replacement of SFA with EVOO cause moderate beneficial cardiometabolic and hepatic effects.
	**Intervention**	SFA diet and EVOO diet (82% of fat replaced by high polyphenol EVOO) Baseline vs. 16 weeks		
	**Technique**	Untargeted and targeted UHPLC-HRMS		
	**Sample**	Plasma and urine		
	**MSI**	Level 3		
	**Statistical analysis**	Univariate		

* Indicates discriminant metabolites down-regulated for the obese group compared with the control group, both treated with HT. BCAAs, branched-chain amino acids; EVOO, extra virgin olive oil; FA, fatty acid; GC, gas chromatography; HCHF, high carbohydrate and high fat diet; HFHF, high fructose and high fat diet; HRMS, high-resolution mass spectrometry; HT, hydroxytyrosol; LC, liquid chromatography; MS, mass spectrometry; MSI, Metabolomics Standards Initiative; NMR, nuclear magnetic resonance spectroscopy; OAHSA, oleic acid hydroxyl stearic acid; OPLS-DA, orthogonal projection to latent structures-discriminant analysis; PC, phosphatidylcholine; PCA, principal component analysis; PE, phosphatidylethanolamine; PLS-DA, partial least square-discriminant analysis; QqTOF, quadrupole-time-of-flight mass spectrometry; TOF, time-of-flight; SFA, saturated fatty acid; UHPLC, ultra-high pressure liquid chromatography; UPLC, ultra-pressure liquid chromatography; UPLC-Q/TOF-MS, ultra-performance liquid chromatography quadrupole/time-of-flight-mass spectrometry.

**Table 4 metabolites-13-00472-t004:** Results of in vitro experimental metabolomics studies.

Reference	Study Data		Main Metabolites Identified after Olive Oil and Its Minor Bioactive Components Consumption	Conclusions
**Fernandez-Arroyo et al., 2012** [47]	**Experimental design**	Colon adenocarcinoma HT29 and SW480) 14 olive oil extracts from EVOO at concentrations of 0.01% and 0.1% for 24 h. Control vs. treated cells	Up-regulated in culture medium: vanillin, 4-OH-benzoic acid, vanillic acid, HT acetate, 10-H-oleuropein aglycone, syringaresinol, acetoxy-pinoresinol, pinoresinol, HT, elenolic acid, luteolin, methyl-decarboxymethyl oleuropein aglycone and apigenin (phenolic compounds). Up-regulated in the cytoplasm: decarboxymethyl oleuropein aglycone, oleuropein aglycone, acetoxy-pinoresinol, elenolic acid, methyl-decarboxymethyl oleuropein aglycone (phenolic compounds) and quercetin, methyl-hydroxy- decarboxymethyl oleuropein aglycone and methyl-luteolin (metabolites)	Association of quercetin and oleuropein aglycone (and its derivatives) with the antiproliferative and pro-apoptotic effect
	**Technique**	Targeted Nano-LC-ESI-TOF-MS		
	**Sample**	Culture medium and cytoplasm		
	**MSI**	Level 3		
	**Statistical analysis**	Univariate		
**Rocchetti et al., 2020** [48] *****	**Experimental design**	In vitro gastrointestinal digestion Five commercial EVOOs were compared	Up-regulated: peonidin, luteolin, pelargonidin, hispidulin (flavonoids), oleuropein, HT (other phenolics), 4-hydroxybenzoic acid (phenolic acids), 2α,7β,15β,18-tetraacetoxy-cholest-5-en-3α-ol (cholesterol analogs), nebrosteroid L (ergosterol derivatives), 6-O-(Glcb)-(25R)-5α-spirostan-3β,6α,23S-triol (spirostanol derivatives)	EVOO in vitro digestion modifies the bioaccessibility of minor bioactive molecules: mainly secoiridoides (oleuropein) and phenolic alcohols (tyrosol and HT), and flavonoids (cyanidin and luteolin)
	**Technique**	Untargeted UHPLC-QTOF		
	**Sample**	Serum		
	**MSI**	Level 3		
	**Statistical analysis**	Multivariate Unsupervised HCA Supervised OPLS-DA		

* Indicate studies of intake metabolites. ESI, electrospray ionization; EVOO, extra virgin olive oil; HCA, hierarchical cluster analysis; HT, hydroxytyrosol; LC, liquid chromatography; MS, mass spectrometry; MSI, Metabolomics Standards Initiative; OPLS-DA, orthogonal projection to latent structures-discriminant analysis; PC, phosphatidylcholine; PCA, principal component analysis; PE, phosphatidylethanolamine; PLS-DA, partial least square-discriminant analysis; QTOF, quadrupole-time-of-flight mass spectrometry; TOF, time-of-flight; UHPLC, ultra-high pressure liquid chromatography.

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
