# Peer review of "Metabolomic-Based Studies of the Intake of Virgin Olive Oil: A Comprehensive Review"

_metabolites, 2023, doi:10.3390/metabo13040472_

Round 1

Reviewer 1 Report

The review is focused on the state of the art regarding the metabolomics studies about olive oil and human health. The argument of the review is interesting and with great potential. Anyway, as also highlighted by the authors, few works have been produced in this field. Moreover, some considered studies also include a specific diet so it is not clear if the metabolite changes are due to diet or olive oil. At the end of the reading, it is not clear if olive oil assumption is responsible for metabolite changes and, most important, it is not clear if the effects are positive or not for the human health prospective. In fact, conclusion section lacks of a final well-defined consideration that can be proposed in the light of this review. A severe lack in the article is the fact that authors refer to virgin olive oil (VOO) in all the text although VOO is just one of the several classes of olive oils and not a term to indicate the whole food matrix. In fact, most of the considered studies have been carried out on extra virgin olive oil (EVOO) and not VOO. The well knowledge of the matrix you are discussing in the paper is the first step for the work.

In the light of all these consideration, the paper can not be accepted in the actual status and major revision are needed before consider it to be published. In particular, it is important to change the discussion and the considerations in order to underline the potential effect of olive oil assumption on the metabolism.

General consideration

In the title and in all the paper you talk about virgin olive oil (VOO) as the “protagonist” but your paper is focused on different olive oil categories (EVOO, VOO,…). For this reason I suggest to replace virgin olive oil (VOO) with olive oil. Virgin olive oil is a class of olive oil, you can’t use it as representative of the whole foodstuff.

All the text

in vitro has to be in italics

in the case studies discussion you indicate that a targeted or untargeted analysis has been used. Specify the type of analysis (NMR, LC, GC,…)

Specific comments

line 88: I suggest to add that the main limit of NMR is its low sensitivity, respect to mass spectroscopy

line 144: in discussing study 26 you talk about VOO at the beginning and in the final part you talk about EVOO. VOO and EVOO are two different classes of olive oil. You have to use to proper one, basing on what used in the study.

line 156: substitute trypto-phan with tryptophan

line 157: substitute How-ever with However

line 166-168: this sentence is very unclear and grammatically bad written

Table 1: reference numbers in the text does not coincide with those in the table. For example, 26, 27, 28, etc. Check the references numbers in all the paper

line 349: “or” or “and”

Author Response

Metabolomic-based studies of the intake of virgin olive oil: a comprehensive review

Reviewer 1

The review is focused on the state of the art regarding the metabolomics studies about olive oil and human health. The argument of the review is interesting and with great potential. Anyway, as also highlighted by the authors, few works have been produced in this field. Moreover, some considered studies also include a specific diet so it is not clear if the metabolite changes are due to diet or olive oil. At the end of the reading, it is not clear if olive oil assumption is responsible for metabolite changes and, most important, it is not clear if the effects are positive or not for the human health prospective. In fact, conclusion section lacks of a final well-defined consideration that can be proposed in the light of this review.

The beneficial effect of VOO and EVOO on health is well known, but there is lack of information about molecular mechanisms of olive oils components. In addition, it is very difficult to separate nutritional effects of one individual dietary component from the complete diet, and in this case, olive oil is usually consumed as a part of the Mediterranean diet, which also contains a great number of healthy nourishments. Therefore, we agree that the results of metabolomic studies with a supplemented Mediterranean diet are not the most useful for attributing a clear effect of its benefits to olive oil, since the Mediterranean diet also has an effect. We have reflected this issue in different sentences in lines 158-159, 178-179 and in the conclusion paragraph in lines 438-442.

A severe lack in the article is the fact that authors refer to virgin olive oil (VOO) in all the text although VOO is just one of the several classes of olive oils and not a term to indicate the whole food matrix. In fact, most of the considered studies have been carried out on extra virgin olive oil (EVOO) and not VOO. The well knowledge of the matrix you are discussing in the paper is the first step for the work.

We agree with the reviewer that the food matrix is a very important factor to evaluate the effect of foods. Both VOO and EVOO are oils exclusively obtained for the cold press of olive fruits, and only differ on the presence of minor compounds that makes EVOO a higher quality oil evaluated by the acidity (caused by free fatty acids) and sensorial properties (related to the minor bioactive compounds). Actually, the food matrix is similar in both and, therefore, its influence would be the same. However, the effects of both olive oils, although similar, may differ because differences in minor components, and for that reason, EVOO is used in the majority of studies, since the amount of bioactive compound should be higher than in VOO. To clarify this issue, we have indicated the type of olive oil used in every selected work.   

In the light of all these considerations, the paper cannot be accepted in the actual status and major revision are needed before consider it to be published. In particular, it is important to change the discussion and the considerations in order to underline the potential effect of olive oil assumption on the metabolism.

As suggested, the discussion has been modified and conclusion has been rewritten, considering all aspect indicated by reviewers

General consideration

In the title and in all the paper you talk about virgin olive oil (VOO) as the “protagonist” but your paper is focused on different olive oil categories (EVOO, VOO,…). For this reason I suggest to replace virgin olive oil (VOO) with olive oil. Virgin olive oil is a class of olive oil, you can’t use it as representative of the whole foodstuff.

The general term olive oil is defined as the mix of virgin olive oil and refined olive oil (https://www.internationaloliveoil.org/olive-world/olive-oil/). Although olive oil is also considered a healthy fat, it does not have the same beneficial health properties since, its content of bioactive phenolic compounds is much lower. Indeed, the beneficial effect of VOO and EVOO on health is well known, but not of refined olive oil or common olive oil. For that reason, we wanted to evaluate the effect of well-known healthy VOO or EVOO,  and decided to indicate VOO in a general way (in the title and introduction sections), specifying the specific oil included in each paper (olive oil, virgin olive oil or extra virgin olive oil) in the results section, in tables and in text.

For example, González-Domínguez et al., 2020, Wang et al., 2018, Ma et al., 2018; Errazuriz et al., 2017; Ma et al., 2017, Fernandez-Arroyo et al., 2012, Mellert et al., 2011 used olive oil. Farras et al., 2022, Fernández-Castillejo et al., 2021; Guasch-Ferre et al., 2020; Yu et al., 2017, used virgin olive oil and Zhi-hao et al., 2022; Ruocco et al., 2022, Rocchetti et al., 2020, Agrawal et al., 2017, Toledo et al., 2017;  Wang et al., 2017; Vázquez-Fresno et al., 2015; Ferreiro-Vela et al., 2013, used extra virgin olive oil.  We reconsidered your comment and the authors finally decided not to change virgin olive oil to olive oil as they are not comparable oils.

All the text

In vitro has to be in italics

As suggested, we have modified “in vitro” in italics in all the text

In the case studies discussion you indicate that a targeted or untargeted analysis has been used. Specify the type of analysis (NMR, LC, GC,…)

As suggested, the type of analysis has been specified for each work through the text.

Specific comments

Line 88: I suggest to add that the main limit of NMR is its low sensitivity, respect to mass spectroscopy

As suggested, we add this limit of NMR in lines 88-89

Line 144: in discussing study 26 you talk about VOO at the beginning and in the final part you talk about EVOO. VOO and EVOO are two different classes of olive oil. You have to use to proper one, basing on what used in the study.

We have indicated EVOO in line 146

Line 156: substitute trypto-phan with tryptophan

We have corrected trypto-phan with tryptophan (line 157)

Line 157: substitute How-ever with However

We have corrected How-ever with However (line 158)

Line 166-168: this sentence is very unclear and grammatically bad written

We have reviewed and modified the sentence (page 4, lines 166-168).

Table 1: reference numbers in the text does not coincide with those in the table. For example, 26, 27, 28, etc. Check the references numbers in all the paper

Sorry for the mistake, we have checked and corrected the references numbers

Line 349: “or” or “and”

As suggested, we have removed “or” (lines 339-340)

Reviewer 2 Report

The work is intersting as it focuses on an emerging tool to address complicated nutritional and health issues. Metabolomic studies are important for all edible categories of olive oil, but mainly to the two highest ones virgin and extra virgin. Few are the queries tha I have.

p.1, l.46. oleuropein is not found in olive oil as it is water soluble and prone to enzymatic hydrolysis. Oleacein is the corresponding hydroxytyrosol derivative as is oleocanthal of tyrosol.

p.2, l.47,    similarly, flavonoid are not among the most important phenols, as these are by far the hydroxytyrosol and tyrsol free and bound forms

In p.11, l.243, maybe the following  reference is relevant for the hydroxytyrosol and tyrosol metabolites

Serreli et al., Molecules. 2021 Dec; 26(24): 7480. Conjugated Metabolites of Hydroxytyrosol and Tyrosol Contribute to the Maintenance of Nitric Oxide Balance in Human Aortic Endothelial Cells at Physiologically Relevant Concentrations

What I miss and i think will strengthen the importance of the review and the discussion is to comment on whether the available data of metabolomic approaches can support (or so far not used and should toward this direction) the health claims that olive oil can officially bear in the bottle. Becasue it is not only the clear one mentioned by EFSA for polyphenols (incorportated into the EC regulation (432/2012)-but also considering the levels in Vitamin E and oleic acid two more that can be applicable to any commodity being a source of vitamin E (Vitamin E contributes to the protection of cells from oxidative stress The claim may be used only for food which is at least a source of vitamin E as referred to in the claim SOURCE OF [NAME OF VITAMIN/S] AND/OR [NAME OF MINERAL/S] as listed in the Annex to Regulation (EC) No 1924/2006) and rich in oleic acid (Oleic acid: Replacing saturated fats in the diet with unsaturated fats contributes to the maintenance of normal blood cholesterol levels. Oleic acid is an unsaturated fat. The claim may be used only for food which is high in unsaturated fatty acids, as referred to in the claim HIGH UNSATURATED FAT as listed in the Annex to Regulation (EC) No 1924/2006.) or generally monounsaturates (Monounsaturated and/or polyunsaturated fatty acids Replacing saturated fats with unsaturated fats in the diet contributes to the maintenance of normal blood cholesterol levels [MUFA and PUFA are unsaturated fats] The claim may be used only for food which is high in unsaturated fatty acids, as referred to in the claim HIGH UNSATURATED FAT as listed in the Annex to Regulation (EC) No 1924/2006.) taking inot account the data considered by EFSA to give the corresponding approvals

Author Response

Metabolomic-based studies of the intake of virgin olive oil: a comprehensive review

Reviewer 2

The work is interesting as it focuses on an emerging tool to address complicated nutritional and health issues. Metabolomic studies are important for all edible categories of olive oil, but mainly to the two highest ones virgin and extra virgin. Few are the queries that I have.

P.1, l.46. Oleuropein is not found in olive oil as it is water soluble and prone to enzymatic hydrolysis. Oleacein is the corresponding hydroxytyrosol derivative as is oleocanthal of tyrosol.

Oleuropein is the secoiridoid glycoside of hydroxytyrosol. Oleuropein is generally the most prominent phenolic compound in olive cultivars and can reach concentrations of up to 140 mg g−1 on a dry matter basis in young olives. Its hydrolysis release hydrotyrosol during the maturation of the fruit. Therefore, the amount of oleuropein depends mainly on the time of ripening and oil preparation. VOO prepared from young fruits at the beginning of the production campaign contents more oleuropeína aglycione and less hydroxityrosol that at the end of the campaign (doi: 10.3797/scipharm.0912-18).  

Therefore, we have modify the text changing “derivatives” with “precursors” to avoid misunderstanding (line 46)

P.2, l.47. Similarly, flavonoids are not among the most important phenols, as these are by far the hydroxytyrosol and tyrsol free and bound forms

Thank you very much for the comment. We have removed this sentence.

In p.11, l.243, maybe the following reference is relevant for the hydroxytyrosol and tyrosol Metabolites

Serreli et al., Molecules. 2021 Dec; 26(24): 7480. Conjugated Metabolites of Hydroxytyrosol and Tyrosol Contribute to the Maintenance of Nitric Oxide Balance in Human Aortic Endothelial Cells at Physiologically Relevant Concentrations

As suggested, we have included Serreli et al, 2021 as reference “37”, in line 253

What I miss and i think will strengthen the importance of the review and the discussion is to comment on whether the available data of metabolomic approaches can support (or so far not used and should toward this direction) the health claims that olive oil can officially bear in the bottle. Becasue it is not only the clear one mentioned by EFSA for polyphenols (incorportated into the EC regulation (432/2012)-but also considering the levels in Vitamin E and oleic acid two more that can be applicable to any commodity being a source of vitamin E (Vitamin E contributes to the protection of cells from oxidative stress The claim may be used only for food which is at least a source of vitamin E as referred to in the claim SOURCE OF [NAME OF VITAMIN/S] AND/OR [NAME OF MINERAL/S] as listed in the Annex to Regulation (EC) No 1924/2006) and rich in oleic acid (Oleic acid: Replacing saturated fats in the diet with unsaturated fats contributes to the maintenance of normal blood cholesterol levels. Oleic acid is an unsaturated fat. The claim may be used only for food which is high in unsaturated fatty acids, as referred to in the claim HIGH UNSATURATED FAT as listed in the Annex to Regulation (EC) No 1924/2006.) or generally monounsaturates (Monounsaturated and/or polyunsaturated fatty acids Replacing saturated fats with unsaturated fats in the diet contributes to the maintenance of normal blood cholesterol levels [MUFA and PUFA are unsaturated fats] The claim may be used only for food which is high in unsaturated fatty acids, as referred to in the claim HIGH UNSATURATED FAT as listed in the Annex to Regulation (EC) No 1924/2006.) taking into account the data considered by EFSA to give the corresponding approvals

Thank you for the reviewer’s interesting suggestion. As indicated in the manuscript, there is still a lack of information about molecular mechanism of olive oil components, and therefore, the available data of metabolomic approaches cannot support olive oil health. We agree with the importance of highlight this information in the text, and we have included the detailed information below into the discussion section. Please, see lines 413-428. We also include additional references. Please, see lines 627-630.

It is worth to mention, that metabolomic analyses, specially, those targeted approaches in which the validation and quantification of specific metabolites are carried out, can be an useful additional tool for supporting health claims, since they can predict health risk or evaluate dietary intake [18]. The increasing of scientific evidence relating to functional ingredients and their health effect might also be interesting for stakeholders and food companies, which could benefit from the added value attributed to their product by the presence of the ingredient responsible for the claimed bioactivity [56]. The EFSA´s health claim criteria for functional foods requires information about the substance (bioactive compound), the study of the physiological effects, and the estimation of the cause-effect relationship. In this lats step, one of the main problems is that many of the studies use biomarkers not significant reliable by the Agency. In this regard, the use of metabolomics could play an important role. However, taking into consideration that nutrimetabolomics is still a young science, under development, and that more standardized and well/designed studies are necessaries [57], we are still far from the implementation of metabolomics as a routinary tool for health claims support. Therefore, at this moment, available data cannot support olive oil EFSA health claims.

(18)     Lioupi, A., Nenadis, N., & Theodoridis, G. (2020). Virgin olive oil metabolomics: A review. Journal of Chromatography B, 1150, 122161.

(56)     Jones, P. J., Asp, N. G., & Silva, P. (2008). Evidence for health claims on foods: how much is enough? Introduction and general remarks. The Journal of Nutrition, 138(6), 1189S-1191S.

(57)     Shibutami, E., & Takebayashi, T. (2021). A Scoping Review of the Application of Metabolomics in Nutrition Research: The Literature Survey 2000–2019. Nutrients, 13(11), 3760.

Reviewer 3 Report

This review is summarizes the available scientific evidence relating to the metabolic effects of virgin olive oil or its bioactive compounds in human, animal and in vitro studies using a metabolomics approach.

The manuscript proposed for expertise is perfectly supported by the many recent references in the field. The methodology used allows readers to fully understand the choice of targeted references. However, authors should better explain their inclusion and exclusion criteria; i) why reject articles not written in English, ii) why reject reviews when they can provide sources of information and above all guidance.

The authors do not only list the references but a discussion is also conducted. Discussion with critical aspects, which is of interest to the reader.

Tables should be redesigned to improve their content.Perhaps it would be preferable to dissociate tables 1 and 2.

It would also be desirable for the authors to add at least 1 to 2 figures instead of writing paragraphs (by examples; 181-208, 289-327),  that are certainly complete but deserve to be based on figures.

A thorough proofreading will allow the authors to correct minor errors such as "in vivo" in "in vivo"....

The conclusion should also better demonstrate the interest of this review and in particular show the perspectives of all this.

Author Response

Metabolomic-based studies of the intake of virgin olive oil: a comprehensive review

Reviewer 3

This review summarizes the available scientific evidence relating to the metabolic effects of virgin olive oil or its bioactive compounds in human, animal and in vitro studies using a metabolomics approach.

The manuscript proposed for expertise is perfectly supported by the many recent references in the field. The methodology used allows readers to fully understand the choice of targeted references. However, authors should better explain their inclusion and exclusion criteria;

  1. why reject articles not written in English

Thank you very much for your comment to improve the work. At the beginning we decided to exclude works not written in English, since it is the language most widely used in scientific literature, and the most accessible to everyone. However, we would like to clarify that the search did not yield papers written in a different language. Therefore, to avoid confusion we have decided to remove this exclusion criteria that do not affect selected papers.

  1. why reject reviews when they can provide sources of information and above all guidance.

We did not include reviewers because we wanted to select specific original results, and reviews usually compile these original data. However, we agree that they provide valuable information and, indeed, we have included some of them in the introduction and discussion sections (i.e. references 1, 3, 5, 15, 40).

The authors do not only list the references but a discussion is also conducted. Discussion with critical aspects, which is of interest to the reader. Tables should be redesigned to improve their content. Perhaps it would be preferable to dissociate tables 1 and 2.

As suggested, we have separated table 1 including postprandial studies in a new table 2, and we have also separated the original table 2, in two new tables, 3 with results for animal studies and table 4 with results for in vitro experimental studies.

It would also be desirable for the authors to add at least 1 to 2 figures instead of writing paragraphs (by examples; 181-208, 289-327), that are certainly complete but deserve to be based on figures.

Thank you very much for your comment. We are not sure the type of figures you mean. In fact, instead of a figure a table will be more beneficial, but we think that would repeat information of previous tables and will make the review boring and tedious for readers.

A thorough proofreading will allow the authors to correct minor errors such as "in vivo" in "in vivo"....

As suggested, we have written “in vivo” and “in vitro” in italics throught the text.

The conclusion should also better demonstrate the interest of this review and in particular show the perspectives of all this.

As suggested, the conclusion has been rewritten to highlight the interest of the review (page 19, lines 430-442): “The present review highlights the need of clinical studies necessary to understand the molecular and metabolic mechanisms of action of VOO components. Although metabolomics studies derived from the consumption of VOO and their metabolic routes are scarce, it has been shown that the intake of VOO causes an increase in derivatives of hydroxytyrosol oleuropeína and oleic acid, such as phosphatide derivatives, which may be used as markers of VOO consumption. In addition, studies have identified possible metabolic pathways related to glycolysis, the tricarboxylic acid cycle, and amino acids metabolism that are modulated by VOO intake, and therefore, may be implicated in the benefits of this healthy oil. However, differences in analytical strategies, the heterogeneity of the experimental designs and interventions, the use of different oils and doses, and the fact that many of the studies carried out in humans evaluate the effect of VOO in the frame of Mediterranean diet, do not allow us to reach specific conclusions on particular metabolites and metabolic pathways affected by VOO intake.”

Round 2

Reviewer 1 Report

The authors responded to all the reviewer comments and made proper changes. The paper is now more clear and well legible. For these reasons, the paper can be accepted in present form

Author Response

Thank you very much for your comments